palaeontology, ecology

Mesozoic marine revolution, escalation, predation, echinoids, drill holes

**Author for correspondence:**
Elizabeth Petsios
e-mail: elizabeth_petsios@baylor.edu

# An asynchronous Mesozoic marine revolution: the Cenozoic intensification of predation on echinoids

Elizabeth Petsios[1], Roger W. Portell[2], Lyndsey Farrar[3], Shamindri Tennakoon[2], Tobias B. Grun[2], Michal Kowalewski[2] and Carrie L. Tyler[3]

[1]Department of Geosciences, Baylor University, One Bear Place #97354, Waco, TX 76798-7354, USA
[2]Florida Museum of Natural History, University of Florida, 1659 Museum Road, Gainesville, FL 32611, USA
[3]Department of Geology and Environmental Earth Science, Miami University, 250 S. Patterson Avenue, Oxford, OH 45056, USA

EP, 0000-0002-7955-7412; RWP, 0000-0002-3306-2115; LF, 0000-0003-1057-9338; ST, 0000-0003-1671-6751; TBG, 0000-0002-1776-0212; MK, 0000-0002-8575-4711; CLT, 0000-0003-0168-2093

Predation traces found on fossilized prey remains can be used to quantify the evolutionary history of biotic interactions. Fossil mollusc shells bearing these types of traces provided key evidence for the rise of predation during the Mesozoic marine revolution (MMR), an event thought to have reorganized global marine ecosystems. However, predation pressure on prey groups other than molluscs has not been explored adequately. Consequently, the ubiquity, tempo and synchronicity of the MMR cannot be thoroughly assessed. Here, we expand the evolutionary record of biotic interactions by compiling and analysing a new comprehensively collected database on drilling predation in Meso-Cenozoic echinoids. Trends in drilling frequency reveal an Eocene rise in drilling predation that postdated echinoid infaunalization and the rise in mollusc-targeted drilling (an iconic MMR event) by approximately 100 Myr. The temporal lag between echinoid infaunalization and the rise in drilling frequencies suggests that the Eocene upsurge in predation did not elicit a coevolutionary or escalatory response. This is consistent with rarity of fossil samples that record high frequency of drilling predation and scarcity of fossil prey recording failed predation events. These results suggest that predation intensification associated with the MMR was asynchronous across marine invertebrate taxa and represented a long and complex process that consisted of multiple uncoordinated steps probably with variable coevolutionary responses.

## 1. Introduction

Taxonomic and ecological diversification during the Mesozoic era, which ultimately gave rise to modern marine ecosystems, was associated with functional innovations and increasingly complex trophic webs, energy budgets and community structures [1–4]. This Mesozoic marine revolution (MMR) is commonly attributed to escalation, or enemy-driven evolution, and is associated with a notable increase in antagonistic trace-producing biotic interactions during the Cretaceous, classically identified as the apex of the MMR [5]. The timing of taxonomic radiations in several marine invertebrate groups has been correlated with increasing frequency of biotic traces [6,7], suggesting a causal link between increasing predation pressure, and the diversification of predator and prey groups involved in an evolutionary arms race [8–10]. However, the role of predation intensification and competition in driving global macroevolutionary trends in prey and predators is controversial (e.g. [11–13]). Furthermore, several studies have pointed to a potentially diachronous intensification across different invertebrate groups and regions [14–19]. Echinoids, an ecologically important clade of marine invertebrates, have been

neglected in these analyses, even though they are well known for their potential for recording biotic interactions in their skeletal remains. Quantifying trends of biotic interactions among echinoids can enhance our understanding of the timing, mode and ultimately the significance of predation intensification and facilitate the identification of any associated escalatory macroevolutionary trends across taxa during the MMR.

Although echinoids are one of the most diverse fossil invertebrate groups in Meso-Cenozoic marine ecosystems, temporal trends in traces of predatory behaviour preserved on echinoid tests have not been assessed adequately. Drill holes are one of the most intensively studied proxies for predation in ancient ecosystems [20,21] because these traces are readily identifiable as biogenic, can often be attributed to specific trace makers and, unlike other forms of predation, do not necessarily result in the complete destruction of the prey skeleton [22,23]. The efforts to systematically quantify the intensity of drilling predation at the population level have focused primarily on traces of predation on molluscs [20,24,25] and, to a lesser extent, brachiopods [26–28]. Echinoids are known to be important prey for several vertebrate and invertebrate predatory groups in modern marine ecosystems ([29–31] and references therein), but only carnivorous cassid gastropods are known to produce drill holes in the process of preying upon echinoids (though parasitism by eulimid gastropods may also produce significantly smaller drill holes). While drilling intensity in fossil echinoid populations has been quantified on a case-by-case basis (e.g. [30,32–38]), a systematic survey across the MMR interval is lacking, limiting our ability to determine the importance of echinoid-targeted predation during the MMR, including its role as a potential community-wide driver of Meso-Cenozoic macroevolutionary trends. Predation and other biotic interactions are also regarded as important drivers of ecological and morphological trends in several clades [39–41]. Starting in the Early Jurassic, echinoids experienced a relatively rapid radiation of infaunal lineages (the Atelostomata and Neognathstomata clades, collectively the 'irregular echinoids') which continued into the early Cenozoic, with infaunal echinoids surpassing the diversity of epifaunal echinoids in the Cretaceous [42,43]. This adaptive radiation has been attributed to colonization and expansion into unoccupied niches [42,44,45], or alternatively, as a response to increasing predation pressure in the Mesozoic [46,47].

Here, we employ a dataset of drill hole traces that includes the extensive systematic surveys of fossil echinoid populations to quantitatively estimate drilling predation pressure on echinoids across the Mesozoic and Cenozoic, encompassing the interval of escalation of biotic interactions associated with the MMR and the adaptive radiation of infaunal echinoids. Echinoids and their cassid predators are very rarely preserved together in fossil assemblages, so the drill hole record offers a means to quantitatively assess the intensity of predation at the population level. The importance of the intensification of antagonistic biotic interactions in marine benthic ecosystems is explored by establishing temporal trends of drilling predation on echinoids relative to mollusc-targeted predatory drilling and in conjunction with the diversification of cassid gastropods and the infaunalization of echinoids.

## 2. Methods

Drill hole frequencies in echinoid populations were tabulated based on surveys of nine museum collections (electronic

supplementary material, table S1) supplemented by field-collected samples acquired across multiple sites in the southeastern USA. Data were assembled as part of an ongoing effort to populate the Echinoid-Associated Traces database (EAT dataset), a developing repository for global data on trace-producing biotic interactions involving echinoids (electronic supplementary material, table S1). Taxa collected from unique localities and stratigraphic units were treated as populations, and only populations with a minimum of 10 individuals were used in drill hole frequency calculations. Middle Jurassic to Holocene echinoid populations were surveyed, and stage-level temporal resolution was determined whenever possible. Field and museum data were supplemented with literature reports of drill hole frequencies in echinoid populations (LIT dataset). A total of 263 populations (201 from the EAT dataset and 62 from the LIT dataset), representing 123 species, 34 823 individuals and spanning 36 chronostratigraphic ages, were used to tabulate drill hole frequencies. Sampled populations spanned at least three continents across all time intervals, and represent echinoid populations from North America, Central America/Caribbean, Eurasia and the Indo-Pacific/Oceania (electronic supplementary material, figure S1). Both epifaunal regular echinoids and epifaunal-to-infaunal irregular echinoid taxa were included. Regular echinoids were underrepresented relative to irregular echinoids in the dataset, likely due to taphonomic bias [48,49], but were nevertheless sampled in all major time intervals. For specimens surveyed in the EAT dataset, only specimens that had greater than 50% of the total test surface area visible were used for drill hole identification, i.e. broken specimens that were missing more than half of the test, or specimens embedded in matrix obscuring more than half of the test surface were not surveyed. The taphonomic grade of each specimen surveyed for the EAT dataset was determined using a semi-quantitative scoring system modified from Nebelsick & Kowalewski [37], in order to test for the effects of the preservation quality of the fossils on our ability to successfully identify biogenic traces. The extent of abrasion on the ambital part of the test, the periproct, the peristome and apical disc was scored separately, and then summed to give an overall taphonomic grade score for each specimen, with more poorly preserved specimens being assigned a higher score (electronic supplementary material, table S2 and Supplemental Methods).

Drill holes were identified and interpreted as predatory in origin based on characteristics that have been traditionally used to diagnose cassid predation in previous studies of both modern and fossil populations [31,34,50–53], such as the relative size of the hole which is generally larger than 0.5 mm, except for traces on minute echinoid taxa; (i.e. [50,51]) and outline shape of the hole including circular, subcircular, irregular, rectangular, elongated or notched holes [30,31,33]. Traces interpreted as non-predatory, such as minute holes attributed to eulimid gastropod parasitism [54], were excluded. A total of 321 drill holes (310 complete; 11 incomplete) attributed to cassid predation were observed in populations surveyed in the EAT dataset. Drill hole frequencies from the literature were included in the LIT dataset if authors specified that drilling frequencies were calculated from population-level surveys and attributed the traces to predation. For fossil populations, chronostratigraphic age was assigned based on the most up-to-date information available regarding the stratigraphic age of the geological units that the fossils were collected or reported from.

Markov chain Monte Carlo (MCMC) time-series models were used to explore and potentially delineate distinct phases in the history of drilling predation. The models simultaneously considered uneven sampling between time bins and heterogeneous variability in population drilling frequencies through time. MCMC models were used to simulate two and three phases of drilling intensification across the studied interval, by resampling with replacement from distributions of pairwise first differences

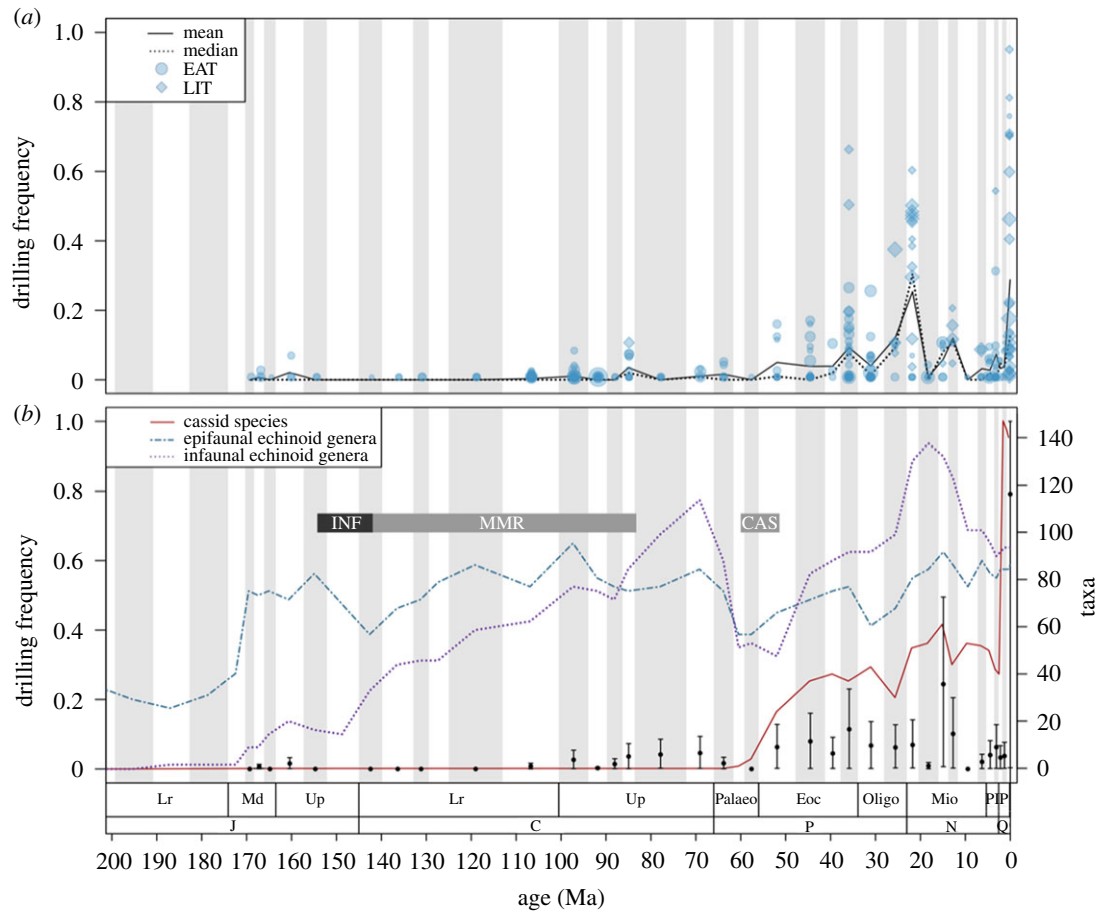

**Figure 1.** Time series showing (a) raw population-level, mean and median drilling frequency in surveyed Jurassic to Holocene echinoids, and (b) sample standardized mean drilling frequencies with 95% confidence intervals, overlain with diversity curves (lines) for epifaunal and infaunal echinoid genera and cassid gastropod species. Point size represents the relative sample size of the sampled population. Circles indicate population drilling frequencies from the EAT dataset, while diamonds represent population drilling frequencies reported from the literature. Grey horizontal bars indicate the timing of the initial radiation of infaunal echinoids (INF), the escalation of mollusc-targeting drilling (MMR), and the initial radiation of cassid gastropods (CAS), as calculated in the changepoint analyses (electronic supplementary material, figure S9) in the case of cassid and infaunal echinoid diversity, and as identified from the literature in the case of the timing of peak MMR. (Online version in colour.)

in drilling frequencies between adjacent time bins. Each MCMC model represented a hypothesis for either two-phase or three-phase trajectory in drilling frequency through time. Two-phase models were constructed with pre- and post-drilling intensification phases, while three-phase models were constructed with a pre-drilling intensification phase, a transitional phase and a post-drilling intensification phase. Within the interval of interest, all possible two-phase and three-phase model phase durations at the level of the chronostratigraphic stage were simulated.

Model fit to the observed drilling frequency data was determined from calculating the sum of squared deviations for each interval (electronic supplementary material, Methods). As drilling frequency data from literature sources (electronic supplementary material, figure S2), may overrepresent highly drilled populations [26], we carried out two analyses to determine the degree to which the addition of literature data may alter analytical outcomes. That is, we calculated two- and three-phase MCMC model fit for both a combined EAT and LIT dataset as well as the systematically sampled EAT dataset only.

## 3. Results

Surveyed Jurassic to Quaternary echinoid populations exhibited increased mean and median drilling frequencies through time, both in the raw population-level drilling frequencies and in sample-standardized mean drilling frequencies

(figure 1). Despite this general increase, most echinoid populations exhibited zero or low drilling frequencies in any given time period (electronic supplementary material, figure S3). The increase in the mean and median population-level drilling frequencies, observed in both the EAT and LIT datasets, was the result of more frequent, but still relatively rare, occurrences of highly drilled populations in the Cenozoic (figure 1; electronic supplementary material, figures S4A and S5A). Specifically, a notable increase in the mean and between-bin first differences in drilling frequency was observed starting in the Eocene and continuing towards the recent, with the highest drilling frequencies occurring in the Holocene populations. A weak but significant partial correlation was observed with drilling frequency in fossil populations and mean taphonomic grade (Pearson's $r = -0.248$, $p < 0.001$) and sample age (Pearson's $r = -0.212$, $p = 0.003$), but no significant partial correlation was observed between drilling frequency and sample size (Pearson's $r = -0.023$, $p = 0.749$; electronic supplementary material, table S3).

Multi-phase MCMC simulations delineated the most likely transition times (i.e. initial intensification) in echinoid-targeted predatory drilling as starting sometime in the early Eocene (Ypresian, midpoint 51.9 Ma) and ending in the late Oligocene (Chattian, midpoint 25.6 Ma; figure 2). The Eocene to Oligocene time interval was identified as the

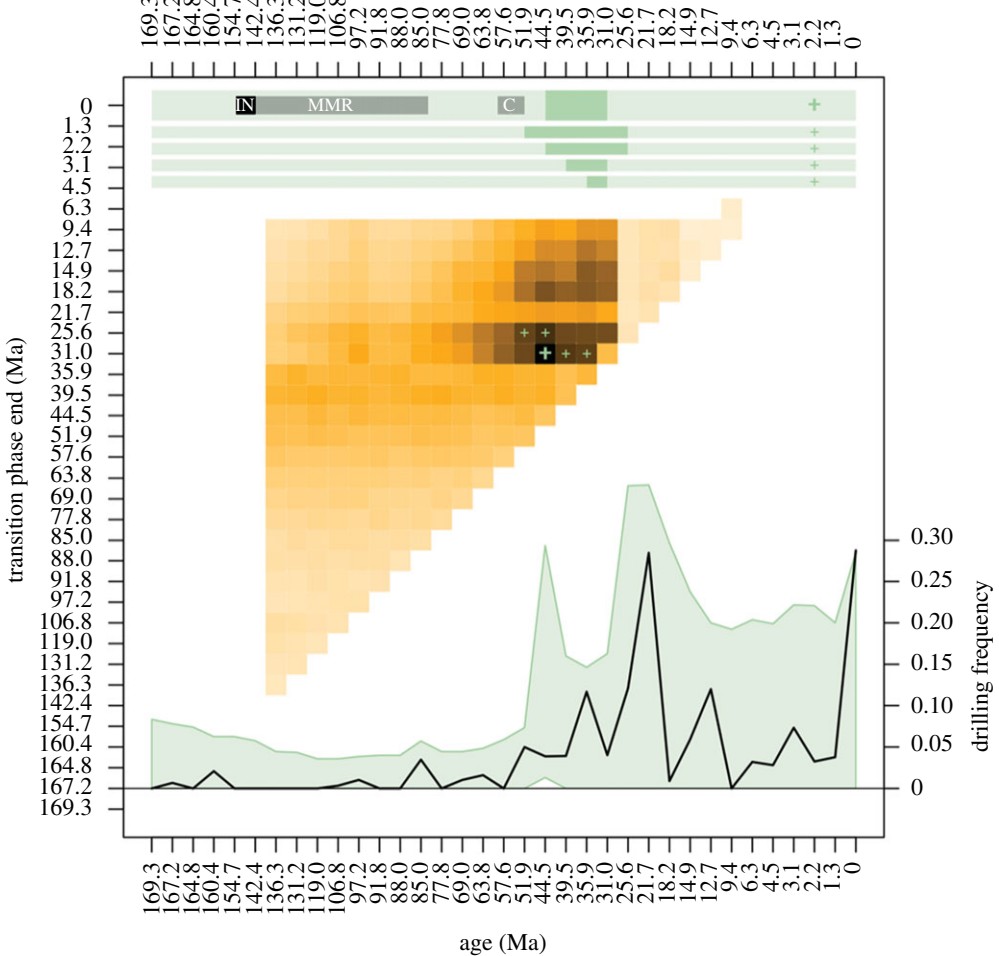

**Figure 2.** Results of three-phase Markov chain Monte Carlo models showing the likely timing of the initiation of drilling intensification in sampled echinoid populations as calculated from the combined EAT + LIT dataset (for the results of the EAT dataset-only analysis, see electronic supplementary material, figure S8). Horizontal bars at the top show five best-fit models (in descending order of model fit from top to bottom), with the duration of transitional phases of elevated drilling represented by a darker-green part of each bar. The transition phase of the best-fit model (bold +) is shown in relation to the timing of the initial radiation of infaunal echinoids (IN), the escalation of mollusc-targeting drilling (MMR) and the initial radiation of cassid gastropods (C), calculated in the changepoint analyses (electronic supplementary material, figure S9). A heatmap of model fit (calculated as the inverse sum of squared deviations) and transition phase age in Ma of every possible modelled transition phase is shown, with dark squares representing modelled transition intervals with the highest model fit. Plus signs (+) correspond to the five best fit models, with the best fit model shown in bold. Raw observed mean population-level drilling frequency is shown at the bottom (black line), with the interquartile confidence bands in drilling frequencies predicted by the best fit model (bold +) shown in green. All model simulations based on 1000 independent iterations. (Online version in colour.)

most likely transition interval in MCMC simulations using both the combined EAT and LIT dataset (figure 2; electronic supplementary material, figure S6) as well as the EAT-only dataset (electronic supplementary material, figures S7 and S8), indicating that this analytical outcome is produced regardless of how data sources are parsed out. Using the combined EAT and LIT dataset, the best fit model ($r = 0.61$) suggests a three-phase history of echinoid-targeting drilling: (i) a pre-intensification phase with invariably low drilling frequencies (Jurassic to early Eocene); (ii) an initial ramp-up phase occurring in the middle Eocene (Lutetian, midpoint 44.5 Ma) and ending in the early Oligocene (Rupelian, midpoint 31 Ma); and (iii) a post-intensification phase encompassing the rest of the Cenozoic when highly drilled populations are more commonplace, yet also highly variable between sampled populations both within and between time bins (electronic supplementary material, figures S4A and S5A). The observed trend in drilling frequency given by the data is consistent with the confidence bands predicted by the best fit model. Moreover, the temporal changes in the

width of confidence bands produced by the model, which simulate uncertainty linked to variable sampling coverage, successfully predicted observed variability in drilling frequencies. In the case of the combined EAT and LIT datasets, all other models with high fit values (the second, third and fourth best fit models) overlapped the predicted timing of the transitional phase of the best-fit model (figure 2), further supporting the most likely intervals of echinoid-targeted drilling intensification as starting in the Eocene.

The timing of the intensification of echinoid-targeted drilling as demonstrated by the best-fit MCMC models was congruent with the timing of the initial radiation of cassid gastropods as estimated from an independent dataset of cassid diversity (electronic supplementary material, table S4). Changepoint analysis [55] pinpointed an initial radiation event beginning in the middle Palaeocene (Selandian, midpoint 60.4 Ma) and ending in the early Eocene (Ypresian, midpoint 51.9 Ma), thus immediately preceding echinoid drilling intensification (electronic supplementary material, figure S9A). Conversely, the timing of the initial radiation of

infaunal echinoids (154 to 142 Ma) and mollusc-targeted drilling (154 to 83 Ma) occurred considerably earlier (electronic supplementary material, figure S9B).

## 4. Discussion

Increasing instances of high drilling frequency in echinoid populations throughout the Meso-Cenozoic were correlated with the appearance and radiation of the proposed trace-makers, cassid gastropods. This temporal coincidence cross-validates the assignment of drill holes observed on fossil echinoids as a likely record of predation by cassid snails. There were a few instances of interpreted drill hole traces reported from the dataset in echinoid populations in the Jurassic and Early Cretaceous that predate the earliest known cassid fossils from the Cenomanian (early Late Cretaceous [56]). While few ($n = 3$), these traces cannot be discounted from the dataset as they meet the criteria for identification as predatory drill holes as discussed in the methods. These traces may have been made by cassids, providing evidence that cassids originated earlier than is currently reported. Alternatively, these traces could represent an extinct or currently unknown echinoid predator that produced traces resembling those made by cassids.

Preservational quality of fossil specimens appears to be weakly correlated with drilling frequency, with samples characterized by more poorly preserved specimens recording generally lower drilling frequencies. Geological age of samples is also weakly correlated with drilling frequency, with older samples recording lower drilling frequencies. However, this correlation does not entirely explain the relationship between preservational quality and drilling frequency, suggesting that poorly preserved specimens may, to a limited extent, impede the preservation or identification of drill holes. This result highlights the importance of systematically recording quantitative or semi-quantitative measures of preservational quality facilitating the assessment of the degree of taphonomic overprinting.

The results support the interpretation that trends in echinoid drilling frequencies throughout the Meso-Cenozoic were a result of increasing predation by cassid gastropods. Generally, diversification of a given taxon is linked to greater population abundance [57] and geographical range [58] of that taxon, suggesting that diversification of cassids probably directly increased drilling predation pressure on echinoid populations throughout this interval. Additionally, cassid gastropods, along with other predatory gastropod groups, experienced a notable increase in maximum body size starting in the Eocene [59], suggesting increased availability and accessibility to their food source. Farrar et al. [31] also report an increase in the diversity of echinoid-associated trace morphologies coincident with the diversification of cassid and eulimid gastropods, suggesting behavioural diversification in echinoid predators and parasites as well.

Epifaunal prey are generally considered more vulnerable to predation [5,60], and epifaunal taxa lacking in other defensive behaviours (e.g. swimming, cementing, spines), often experience higher rates of extinction [61]. Coincident with the rise of durophagous and mollusc-targeted drilling predation during the MMR, the ecological and taxonomic replacement of the previously dominant epifaunal, sessile or slow-moving taxa with infaunal, mobile or cemented

taxa occurred in several marine groups [62,63]. This trend towards infaunalization was also apparent in echinoids, with the Jurassic appearance and later Mesozoic radiation of infaunal echinoids. By the Cretaceous, the diversity of infaunal echinoids exceeded the diversity of epifaunal echinoids. The Cretaceous infaunalization shift is viewed as one of the most important evolutionary events in the history of the group [44,46]. In addition to the predator-avoidance strategies associated with an infaunal lifestyle, it has been postulated that many of the morphological changes associated with infaunalization in Mesozoic echinoids, such as flattening of the test, an increase in ambulacral plate compounding and sinking of the ambulacral petals, are also anti-predatory adaptations [46]. Consequently, as was the case for many other invertebrate groups, the intensification of predation pressure associated with the MMR could have been a significant driving mechanism behind the evolutionary success of the infaunal echinoids. However, here we demonstrate that a significant increase in drilling predation post-dated infaunalization and coincided with the Eocene diversification of the proposed tracemakers, suggesting that cassid predation was unlikely to play major role in driving echinoid infaunalization. Either infaunalization was not driven by predation or other echinoid predators, such as crustaceans or fish, were responsible. Several studies report evidence of durophagous or whole-test ingestion predation on echinoids from the Jurassic and Early Cretaceous [64,65], and the oldest reported evidence of predation on echinoids is from fish regurgitates from the Middle Jurassic [45]. As in most prey groups, however, systematically assessing and quantifying the intensity of durophagy in fossil echinoid populations remains problematic due to the likely destruction of the test during such interactions [66]. Cassid predation pressure, though uncorrelated with echinoid diversification patterns in the Mesozoic, may have contributed to the continued radiation of infaunal echinoid clades, such as clypeasteroids, in the Cenozoic [42], given the high rates of cassid predation in some modern echinoid populations [30,37,67–69].

The intensification of drilling predation of molluscs, in contrast, is highly correlated with trends of infaunalization in mollusc prey taxa [70,71]. The initiation of intense mollusc-targeting drilling in the Cretaceous is contemporaneous with other evidence of intense predation pressure on molluscs, such as increased frequency of repair scars [72], and appears to have driven significant coevolutionary responses in molluscs across the Mesozoic [73]. Echinoid-targeting drilling in the Eocene significantly postdates these escalatory responses in molluscs. This disparity in timing and the lack of a reciprocal response in diversification patterns to the intensification of drilling in echinoids suggests that drilling predators preying on echinoids were not as potent an evolutionarily force as drilling predators targeting mollusc prey. In part, this could be explained by the apparent rarity of echinoid populations that suffered high mortality from drilling predation, even after the Eocene. While some Neogene and especially Recent echinoid populations record high drilling frequencies (i.e. greater than 95% [69]), average population drilling frequency is still low throughout most of the studied interval, suggesting that cassid drilling alone might not have exerted a strong enough selective pressure to produce significant coevolutionary responses in echinoid diversification patterns. The lack of correlation between drilling frequency and echinoid diversification, especially in

the post-intensification interval, suggests that instances of high cassid predation pressure are too sporadic and geographically dispersed to produce strong enough selective pressures to affect global echinoid diversity patterns. Additionally, the low proportion of incomplete drill holes observed throughout the studied interval (less than 4% of all drill holes recorded across all echinoid species) suggests that when cassids do attack, they are highly successful. This high fatality rate, regardless of echinoid life habit, also points to weak selective pressure exerted by cassids on its echinoid prey.

Though echinoids are abundant and diverse prey in Modern food webs, we demonstrate here that the evolutionary history of echinoid-targeting drilling predation, the timing of its intensification and its macroevolutionary consequences on echinoid clades is distinct from that observed in molluscs. The Cenozoic intensification of predation observed here is consistent with a post-Mesozoic component to the MMR, and lends further support to the idea that the MMR was not a single synchronized ecosystem-wide event, but instead represented a series of asynchronous processes with variable coevolutionary significance across different clades of prey and their predators.

Data accessibility. The dataset used in this analysis available from the Dryad Digital Repository: https://doi.org/10.5061/dryad.wstqjq2kn [74].

Authors' contributions. Fieldwork was conducted by E.P., R.W.P., L.F., S.T., M.K. and C.L.T. Sample processing was conducted by E.P., R.W.P., L.F. and S.T. Data analysis was conducted by E.P., M.K. and C.L.T. Additional data were contributed by T.B.G. E.P. wrote the manuscript, with intellectual input and comments from T.B.G., M.K. and C.L.T.

Competing interests. We declare we have no competing interests.

Funding. This work was supported by a National Science Foundation grant to C.L.T. and M.K. (EAR SGP-1630475 and EAR SGP-1630276).

Acknowledgements. We thank the late Anne Molineux, George Phillips, Wally Ward, Dallas Paleontological Society, Roger and Linda Farish, Carmi Milagros Thompson, Sean Roberts and Matthew Dunkelberger for either assistance in the field, collections or specimen imaging. We also thank Katherine Bartels, Jessica Goldstein, Wrik Chatterjee, Tasha Anderson, Sarah Emrick, Erin Graves and the Florida Museum volunteers who assisted with data collection, as well as the Paleobiology Database. In addition, we thank the following people, who contributed 75% of the data downloaded from the PBDB: Austin Hendy, Pete Wagner, Sabine Nürnberg, Matthew Kosnik and Katherine Bulinski. Support for L.F. was provided by the Dry Dredgers, The Paleontological Society, Miami University Graduate School, Miami University Graduate Student Association and the Florida Museum of Natural History. We thank Dr Geerat Vermeij and one anonymous reviewer for their helpful comments and suggestions.

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
