## [Peer Review File · Proceedings of the Royal Society B: Biological Sciences]

Review History

RSPB-2020-1986.R0 (Original submission)

Review form: Reviewer 1 (Geerat Vermeij)

Recommendation

Major revision is needed (please make suggestions in comments)

Scientific importance: Is the manuscript an original and important contribution to its field?

Acceptable

General interest: Is the paper of sufficient general interest?

Good

Quality of the paper: Is the overall quality of the paper suitable?

Marginal

Is the length of the paper justified?

No

Should the paper be seen by a specialist statistical reviewer?

Yes

Do you have any concerns about statistical analyses in this paper? If so, please specify them explicitly in your report.

No

It is a condition of publication that authors make their supporting data, code and materials available - either as supplementary material or hosted in an external repository. Please rate, if applicable, the supporting data on the following criteria.

Is it accessible?

Yes

Is it clear?

Yes

Is it adequate?

Yes

Do you have any ethical concerns with this paper?

No

Comments to the Author

The data here are interesting and very much worth publishing, but I have substantial problems with how these data are interpreted. I should admit here that I do not understand the statistical methods used, and therefore cannot comment on the validity of the statistical conclusions.

The premise of the paper is that there is a post-Mesozoic component to the MMR, a conclusion I agree with and have expressed myself on many occasions. But the authors do not provide any evidence for their particular conclusion. In particular, I note that only about one per cent of the individuals examined were drilled, indicating that cassids are unimportant for most populations as a source of predation. It is unclear if any of the drill holes were unsuccessful; if they were all successful, then cassids are very unlikely to have been selective agents with respect to echinoids.

At the moment, the claim of "coevolution" is based on diversity trends, but these are inherently correlational and not causal. If cassids were important as selective agents, one would have to demonstrate either unsuccessful drilling or other mechanisms by which urchins defend themselves, such as by spines (which probably do not work well on cassids) or escape. I am unaware of any functional studies of urchins in relation to cassid predation despite quite a few studies in which cassid behavior is documented.

The authors make no mention of the huge size increase in cassids after the Eocene. Is there any indication from the echinoids that larger individuals were drilled after the Eocene? Do the authors have any data of urchin drilling from sites or times where cassids do not occur? I note from my experience that cassids are never very common, at least in the shallow-water habitats I have worked in. Can the authors give a percentage of the urchin species that were undrilled, and is there any indication of how those urchins might differ from those that were drilled frequently?

In short, the hypothesis of escalation hinges on characteristics of the species in question and on unsuccessful predation as inferred in fossils. There are many fascinating avenues with which to pursue these question in the cassid-echinoid dynamic, but the present paper does not deliver. - G. J. Vermeij

Review form: Reviewer 2

Recommendation

Major revision is needed (please make suggestions in comments)

Scientific importance: Is the manuscript an original and important contribution to its field?

Good

General interest: Is the paper of sufficient general interest?

Good

Quality of the paper: Is the overall quality of the paper suitable?

Good

Is the length of the paper justified?

Yes

Should the paper be seen by a specialist statistical reviewer?

No

Do you have any concerns about statistical analyses in this paper? If so, please specify them explicitly in your report.

Yes

It is a condition of publication that authors make their supporting data, code and materials available - either as supplementary material or hosted in an external repository. Please rate, if applicable, the supporting data on the following criteria.

Is it accessible?

Yes

Is it clear?

Yes

Is it adequate?

Yes

Do you have any ethical concerns with this paper?

No

Comments to the Author

Overall, this is an interesting, well-written and well-argued paper that is of broad paleobiological interest. However, there is one critical issue that is problematic and most comments that follow are devoted to that.

These comments are based in part on some analyses of the data (S1) provided by the authors.

The problematic issue is the authors' combining two data sets into one, and performing quantitative analyses, central to the results of study, only on the combined data. The two data sets are (1) museum- and collections-derived data (EAT), and (2) literature-derived data (LIT). Each of the two sets provides some interesting insights into the timing of intensification of drilling predation on echinoids, perhaps complementary insights, but the data differ in substantive/significant ways (see Table Rev below). The authors offer no explicit discussion of (a) how these data differ from each other, (b) why they differ in these substantive ways, and, (c) why, given these significant differences, treating them together in the quantitative analyses is

justified.

Table Rev: Some metrics for the two data sets and the combined data as calculated from data provided:

	specimens	pops drilled	DFmean	DFmed		Czpops	CZpops	
	n	n	n					%
Combined	34823	263	2841	0.08	0.01	186	0.71	
EAT	9315		201	321	0.02	0.00	125	0.62
LIT	25508		62	2520	0.26	0.15	61	0.98

The LIT and EAT data differ significantly from each other in almost every metric, especially in DFs (mean DFs differ by an order of magnitude). The differences in DFs between EAT and LIT may arise because LIT data are very much non-random with respect to DF, with researchers (with exception of Grun 2014) publishing results only for populations with DF>0, whereas EAT data are more or less random with respect to DF (presence/absence of drilled specimens, and DF values, was not a criterion for EAT collections, though it is possible that in some museum collections drilled specimens were removed by specialists for further study, although that seems highly unlikely and the authors would probably be aware of this). As a consequence, populations with DF=0, are counted (and very common, 123 of 201 pops) in EAT but only one instance is reported in LIT (Grun 2014).

Given the differences, should these data sets be combined for the analyses of the timing of drilling intensification? Statistically, I do not know how this can be justified. No procedure that I am aware of (resampling, differencing, etc.) can be used to somehow negate or equalize the very different biases of the two data sets. Even if the authors can answer these questions, it would be interesting to analyze EAT and LIT separately and see the congruence (or lack of congruence) in the results. In that context, the methods employed by the authors seem perfectly appropriate for EAT, and it is those analyses that would be the most persuasive because of the non-random nature of the EAT data. Treating the LIT and EAT independently and analyzing each separately, could strengthen the conclusions of the study, namely, that the increase in DF in echinoids was coincident with the diversification of cassids. If analyses of EAT and LIT produce different answers about the timing of the intensification, that would be of interest as well, as would a comparison of EAT and LIT results to the “combined” result.

Additional comments (*italicized*)

There are some minor discrepancies in the text of the ms.

Comment 1.

Text lines 85-89: the authors mentioned two separate tables Table S1 (EAT) and Table S2 (literature). I assume these were merged into Table S1 in the Supp. as Table S2 in Supp is something different

Comment 2.

*Line 93: “A total of 263 populations (198 from EAT and 65 from literature sources)”
263 are listed in the S1 but 201 are EAT (not 198) and 62 LIT (not 65).*

Comment 3.

Figure S1 – “Paleogeographic maps showing the geographic extent of EAT Database Project sampling”

This map clearly includes data from LIT and EAT and not just the latter. This should be made explicit.

Decision letter (RSPB-2020-1986.R0)

10-Sep-2020

Dear Dr Petsios:

I am writing to inform you that your manuscript RSPB-2020-1986 entitled "An Asynchronous Mesozoic Marine Revolution: The Cenozoic Escalation of Predation on Echinoids" has, in its current form, been rejected for publication in Proceedings B.

This action has been taken on the advice of referees, who have recommended that substantial revisions are necessary. With this in mind we would be happy to consider a resubmission, provided the comments of the referees are fully addressed. However please note that this is not a provisional acceptance. The reviewers and editor are interested in the paper but they question some major aspects of the science that must be more convincingly addressed in the revised MS.

Please note that this decision may (or may not) have taken into account confidential comments.

In your revision process, please take a second look at how open your science is; our policy is that **ALL** (maximally inclusive) data involved with the study should be made openly accessible, fully enabling re-use, replication and transparency-- see:

<https://royalsociety.org/journals/ethics-policies/data-sharing-mining/>

Insufficient sharing of data can delay or even cause rejection of a paper.

Full data and code/scripts to enable reuse/replication/repurposing are what this policy intends.

Sincerely,

Dr John Hutchinson, Editor

Associate Editor

Board Member: 1

Comments to Author:

Both of the reviewers believe the paper needs major revision. The comments are very critical. For example, Rev 1 believes that the interpretations do not match the data and Rev 2 believes there is a statistical problem with combining the two data sets analysed in the paper. These are very substantive issues and would ordinarily be grounds for rejection as scientifically unsound. I am intrigued, however, (and swayed) by the fact that each individual reviewer called not for rejection, but major revision, so I have opted to give the authors another chance – each of these same two reviewers would need to have another look at any revised ms.

Reviewer(s)' Comments to Author:

Referee: 1

Comments to the Author(s)

The data here are interesting and very much worth publishing, but I have substantial problems with how these data are interpreted. I should admit here that I do not understand the statistical methods used, and therefore cannot comment on the validity of the statistical conclusions.

The premise of the paper is that there is a post-Mesozoic component to the MMR, a conclusion I agree with and have expressed myself on many occasions. But the authors do not provide any evidence for their particular conclusion. In particular, I note that only about one per cent of the individuals examined were drilled, indicating that cassids are unimportant for most populations as a source of predation. It is unclear if any of the drill holes were unsuccessful; if they were all successful, then cassids are very unlikely to have been selective agents with respect to echinoids.

At the moment, the claim of "coevolution" is based on diversity trends, but these are inherently correlational and not causal. If cassids were important as selective agents, one would have to demonstrate either unsuccessful drilling or other mechanisms by which urchins defend themselves, such as by spines (which probably do not work well on cassids) or escape. I am unaware of any functional studies of urchins in relation to cassid predation despite quite a few studies in which cassid behavior is documented.

The authors make no mention of the huge size increase in cassids after the Eocene. Is there any indication from the echinoids that larger individuals were drilled after the Eocene? Do the authors have any data of urchin drilling from sites or times where cassids do not occur? I note from my experience that cassids are never very common, at least in the shallow-water habitats I have worked in. Can the authors give a percentage of the urchin species that were undrilled, and is there any indication of how those urchins might differ from those that were drilled frequently?

In short, the hypothesis of escalation hinges on characteristics of the species in question and on unsuccessful predation as inferred in fossils. There are many fascinating avenues with which to pursue these questions in the cassid-echinoid dynamic, but the present paper does not deliver. - G. J. Vermeij

Referee: 2

Comments to the Author(s)

Overall, this is an interesting, well-written and well-argued paper that is of broad paleobiological interest. However, there is one critical issue that is problematic and most comments that follow are devoted to that.

These comments are based in part on some analyses of the data (S1) provided by the authors.

The problematic issue is the authors' combining two data sets into one, and performing quantitative analyses, central to the results of study, only on the combined data. The two data sets are (1) museum- and collections-derived data (EAT), and (2) literature-derived data (LIT). Each of the two sets provides some interesting insights into the timing of intensification of drilling predation on echinoids, perhaps complementary insights, but the data differ in

substantive/significant ways (see Table Rev below). The authors offer no explicit discussion of (a) how these data differ from each other, (b) why they differ in these substantive ways, and, (c) why, given these significant differences, treating them together in the quantitative analyses is justified.

Table Rev: Some metrics for the two data sets and the combined data as calculated from data provided:

	specimens	pops	drilled	DFmean	DFmed	Czpops	CZpops
	n	n	n	%			
Combined	34823	263	2841	0.08	0.01	186	0.71
EAT	9315	201	321	0.02	0.00	125	0.62
LIT	25508	62	2520	0.26	0.15	61	0.98

The LIT and EAT data differ significantly from each other in almost every metric, especially in DFs (mean DFs differ by an order of magnitude). The differences in DFs between EAT and LIT may arise because LIT data are very much non-random with respect to DF, with researchers (with exception of Grun 2014) publishing results only for populations with DF>0, whereas EAT data are more or less random with respect to DF (presence/absence of drilled specimens, and DF values, was not a criterion for EAT collections, though it is possible that in some museum collections drilled specimens were removed by specialists for further study, although that seems highly unlikely and the authors would probably be aware of this). As a consequence, populations with DF=0, are counted (and very common, 123 of 201 pops) in EAT but only one instance is reported in LIT (Grun 2014).

Given the differences, should these data sets be combined for the analyses of the timing of drilling intensification? Statistically, I do not know how this can be justified. No procedure that I am aware of (resampling, differencing, etc.) can be used to somehow negate or equalize the very different biases of the two data sets. Even if the authors can answer these questions, it would be interesting to analyze EAT and LIT separately and see the congruence (or lack of congruence) in the results. In that context, the methods employed by the authors seem perfectly appropriate for EAT, and it is those analyses that would be the most persuasive because of the non-random nature of the EAT data. Treating the LIT and EAT independently and analyzing each separately, could strengthen the conclusions of the study, namely, that the increase in DF in echinoids was coincident with the diversification of cassids. If analyses of EAT and LIT produce different answers about the timing of the intensification, that would be of interest as well, as would a comparison of EAT and LIT results to the “combined” result.

Additional comments (italicized)

There are some minor discrepancies in the text of the ms.

Comment 1.

Text lines 85-89: the authors mentioned two separate tables Table S1 (EAT) and Table S2 (literature). I assume these were merged into Table S1 in the Supp. as Table S2 in Supp is something different

Comment 2.

*Line 93: “A total of 263 populations (198 from EAT and 65 from literature sources)”
263 are listed in the S1 but 201 are EAT (not 198) and 62 LIT (not 65).*

Comment 3.

Figure S1 – “Paleogeographic maps showing the geographic extent of EAT Database Project sampling”

This map clearly includes data from LIT and EAT and not just the latter. This should be made explicit.

Author's Response to Decision Letter for (RSPB-2020-1986.R0)

See Appendix A.

RSPB-2021-0400.R0

Review form: Reviewer 1 (Geerat Vermeij)

Recommendation

Accept as is

Scientific importance: Is the manuscript an original and important contribution to its field?

Good

General interest: Is the paper of sufficient general interest?

Excellent

Quality of the paper: Is the overall quality of the paper suitable?

Good

Is the length of the paper justified?

Yes

Should the paper be seen by a specialist statistical reviewer?

No

Do you have any concerns about statistical analyses in this paper? If so, please specify them explicitly in your report.

No

It is a condition of publication that authors make their supporting data, code and materials available - either as supplementary material or hosted in an external repository. Please rate, if applicable, the supporting data on the following criteria.

Is it accessible?

Yes

Is it clear?

Yes

Is it adequate?

Yes

Do you have any ethical concerns with this paper?

No

Comments to the Author

The authors have largely answered my previous queries and have cleared up some of their earlier confusions and misconceptions. I still would have liked a more granular analysis of the data, including instances where echinoids were not drilled and potential cases where cassids occur but their drill holes do not.

Review form: Reviewer 2

Recommendation

Accept with minor revision (please list in comments)

Scientific importance: Is the manuscript an original and important contribution to its field?

Good

General interest: Is the paper of sufficient general interest?

Good

Quality of the paper: Is the overall quality of the paper suitable?

Good

Is the length of the paper justified?

Yes

Should the paper be seen by a specialist statistical reviewer?

No

Do you have any concerns about statistical analyses in this paper? If so, please specify them explicitly in your report.

No

It is a condition of publication that authors make their supporting data, code and materials available - either as supplementary material or hosted in an external repository. Please rate, if applicable, the supporting data on the following criteria.

Is it accessible?

Yes

Is it clear?

Yes

Is it adequate?

Yes

Do you have any ethical concerns with this paper?

No

Comments to the Author

Figure S1: The "numbering" on the panels (D = Cretaceous) and the caption (A=Cretaceous) is reversed.

Figure S2 caption. "This is likely due to both disproportionately greater reporting of highly drilled populations from the primary literature (ref. 1), and also potentially due to the literature data predominantly representing Neogene and Quaternary populations, when drilling frequency is expected to be relatively higher."

The above is undoubtedly correct but a few additional comments may be in order:

- Are the two distributions statistically different? (Chi Squared or some other test)
- The Lit data contain only a single population without drilled echinoids (Grun et al. 2014 Historical Biology), whereas in EAT, those predominate (>50%). So can the absence of such reports in the Lit data be biologically real? That's not likely. Therefore, not only does LIT appear biased towards the high frequencies, but it is also biased against 0 frequencies. I think this is

worth pointing out, even if the MCMC simulations using both the combined EAT and LIT dataset as well as the EAT-only dataset produced similar results.

Decision letter (RSPB-2021-0400.R0)

01-Mar-2021

Dear Dr Petsios

I am pleased to inform you that your manuscript RSPB-2021-0400 entitled "An Asynchronous Mesozoic Marine Revolution: The Cenozoic Intensification of Predation on Echinoids" has been accepted for publication in Proceedings B. Congratulations!!

The referee(s) have recommended publication, but also suggest some minor revisions to your manuscript. Therefore, I invite you to respond to the referee(s)' comments and revise your manuscript. Because the schedule for publication is very tight, it is a condition of publication that you submit the revised version of your manuscript within 7 days. If you do not think you will be able to meet this date please let us know.

Please ensure data accessibility is made crystal clear in your MS-- from <https://www.eat-project.org/> it is still not immediately evident where the dataset and any associated code from this study is located and if it is 100% public.

Sincerely,

Dr John Hutchinson, Editor

Reviewer(s)' Comments to Author:

Referee: 1

Comments to the Author(s).

The authors have largely answered my previous queries and have cleared up some of their earlier confusions and misconceptions. I still would have liked a more granular analysis of the data, including instances where echinoids were not drilled and potential cases where cassids occur but their drill holes do not.

Referee: 2

Comments to the Author(s).

Figure S1: The "numbering" on the panels (D = Cretaceous) and the caption (A=Cretaceous) is reversed.

Figure S2 caption. "This is likely due to both disproportionately greater reporting of highly drilled populations from the primary literature (ref. 1), and also potentially due to the literature data predominantly representing Neogene and Quaternary populations, when drilling frequency is expected to be relatively higher."

The above is undoubtedly correct but a few additional comments may be in order:

- a. Are the two distributions statistically different? (Chi Squared or some other test)
- b. The Lit data contain only a single population without drilled echinoids (Grun et al. 2014 Historical Biology), whereas in EAT, those predominate (>50%). So can the absence of such reports in the Lit data be biologically real? That's not likely. Therefore, not only does LIT appear biased towards the high frequencies, but it is also biased against 0 frequencies. I think this is worth pointing out, even if the MCMC simulations using both the combined EAT and LIT dataset as well as the EAT-only dataset produced similar results.

Author's Response to Decision Letter for (RSPB-2021-0400.R0)

See Appendix B.

Decision letter (RSPB-2021-0400.R1)

09-Mar-2021

Dear Dr Petsios

I am pleased to inform you that your manuscript entitled "An Asynchronous Mesozoic Marine Revolution: The Cenozoic Intensification of Predation on Echinoids" has been accepted for publication in Proceedings B.

Your article has been estimated as being 8 pages long. Our Production Office will be able to confirm the exact length at proof stage.

Data Accessibility section

Open Access

Paper charges

Sincerely,

Proceedings B

Appendix A

Dear Dr Petsios:

I am writing to inform you that your manuscript RSPB-2020-1986 entitled "An Asynchronous Mesozoic Marine Revolution: The Cenozoic Escalation of Predation on Echinoids" has, in its current form, been rejected for publication in Proceedings B.

This action has been taken on the advice of referees, who have recommended that substantial revisions are necessary. With this in mind we would be happy to consider a resubmission, provided the comments of the referees are fully addressed. However please note that this is not a provisional acceptance. The reviewers and editor are interested in the paper but they question some major aspects of the science that must be more convincingly addressed in the revised MS.

Please note that this decision may (or may not) have taken into account confidential comments.

In your revision process, please take a second look at how open your science is; our policy is that *ALL* (maximally inclusive) data involved with the study should be made openly accessible, fully enabling re-use, replication and transparency-- see: <https://royalsociety.org/journals/ethics-policies/data-sharing-mining/>
Insufficient sharing of data can delay or even cause rejection of a paper.
Full data and code/scripts to enable reuse/replication/repurposing are what this policy intends.

Sincerely,

Dr John Hutchinson, Editor
mailto: proceedingsb@royalsociety.org

Dear Dr. John Hutchinson,

Please find attached our revised submission to *Proceedings of the Royal Society Biology*, titled “An asynchronous Mesozoic Marine Revolution: the Cenozoic intensification of predation on echinoids” (**RSPB-2020-1986**), as well as our responses to referee comments. We have addressed the referees concerns regarding the combining of sources to compile the data used for analyses and the interpretations made. Below, we summarize the specific actions we took to address the referee’s comments, and have attached additional detailed responses to specific comments below that. We provide a copy of the referee’s original comments in black text and our responses in red.

To address Dr. Vermeij’s (Referee 1) comments, we have modified the title to replace the word “Escalation” with “Intensification”, and have added a short discussion to clarify the difference. As Dr. Vermeij pointed out, a co-evolutionary response to intensification of drilling predation was either absent (in the case of diversity patterns in the infaunal echinoid clade) or not addressed with our existing data directly. We have clarified the terminology throughout the manuscript to make it clear that we are only referring to the observed intensification of drilling predation targeting echinoids as it correlated to diversity patterns in the groups of interest. We have added points of discussion to the Discussion section focusing on the role (or lack thereof) of selection pressure from cassid gastropods on echinoid populations across the studied interval, and explore potential explanations for a lack of response in echinoid diversification patterns.

To address Referee 2’s concern that inclusion of the LIT dataset skews the results of the analysis due to biased reporting of highly drilled populations in the primary literature, we have re-run the MCMC analysis using only the EAT dataset, and find that our ability to delineate the timing of initial drilling intensification in the Eocene-Oligocene, and thus the crux of our interpretations, was not affected. We have therefore included both results from the MCMC model simulations generated using the combined EAT and LIT dataset, as well as the EAT-only dataset, and provided the results of the separated LIT and EAT data analyses in the supplemental materials. We have clarified the differences between the EAT and LIT database throughout the Methods and Results sections, and have modified figures and figure captions for greater transparency as to the source of the data.

We hope that these changes address the concerns of both the referees and the editorial board.

Sincerely,
Elizabeth Petsios, Ph.D

Referee: 1

Comments to the Author(s)

The data here are interesting and very much worth publishing, but I have substantial problems with how these data are interpreted. I should admit here that I do not understand the statistical methods used, and therefore cannot comment on the validity of the statistical conclusions.

The premise of the paper is that there is a post-Mesozoic component to the MMR, a conclusion I agree with and have expressed myself on many occasions. But the authors do not provide any evidence for their particular conclusion. In particular, I note that only about one per cent of the individuals examined were drilled, indicating that cassids are unimportant for most populations as a source of predation. It is unclear if any of the drill holes were unsuccessful; if they were all successful, then cassids are very unlikely to have been selective agents with respect to echinoids.

At the moment, the claim of "coevolution" is based on diversity trends, but these are inherently correlational and not causal. If cassids were important as selective agents, one would have to demonstrate either unsuccessful drilling or other mechanisms by which urchins defend themselves, such as by spines (which probably do not work well on cassids) or escape. I am unaware of any functional studies of urchins in relation to cassid predation despite quite a few studies in which cassid behavior is documented.

The authors make no mention of the huge size increase in cassids after the Eocene. Is there any indication from the echinoids that larger individuals were drilled after the Eocene? Do the authors have any data of urchin drilling from sites or times where cassids do not occur? I note from my experience that cassids are never very common, at least in the shallow-water habitats I have worked in. Can the authors give a percentage of the urchin species that were undrilled, and is there any indication of how those urchins might differ from those that were drilled frequently?

In short, the hypothesis of escalation hinges on characteristics of the species in question and on unsuccessful predation as inferred in fossils. There are many fascinating avenues with which to pursue these questions in the cassid-echinoid dynamic, but the present paper does not deliver. - G. J. Vermeij

We agree that our initial use of the term "escalation" throughout the manuscript was imprecise. We have thus amended the language throughout the manuscript to more accurately reflect that what we are exploring is the "intensification" of predation. We agree that to demonstrate escalation temporally synchronous co-evolutionary responses (be it morphological, behavioral, ecological, or other) would need to be observed in both predator and prey clades. We partially explore this concept by considering the timing of diversification of the infaunal echinoid clade as it pertains to the timing of intensification of cassid predation, but do not find evidence for escalation in this case.

The above issue does not nullify the key result of our study. We provide extensive empirical support for the claim that the intensification of predation during the MMR is not synchronous across major prey groups, and that cassids, widely perceived as significant echinoid predators, were not associated with the infaunalization of echinoids. The relationship between cassids and echinoids demonstrates that this particular predator-prey relationship intensifies later than the initial MMR "peak" observed in mollusks. As our dataset does not address changes in, for example, predator and prey size, nor changes in stereotypy of traces, we cannot directly test for escalation in this regard and therefore cannot rule these out as potential co-evolutionary responses across our studied interval.

As the reviewer points out, this lack of co-evolutionary response (in biodiversity patterns) could potentially be attributed to the relatively low frequency of cassid drilling in echinoid populations, even after the initial intensification of this behavior in the Eocene, meaning that predation pressure imposed by cassids is potentially too weak to produce an evolutionary response in echinoids. In post-intensification intervals (Neogene and Quaternary) we observe rare and sporadic occurrences of highly drilled echinoid populations, suggesting that even though some populations experience extreme cassid predation pressure, this is too geographically sporadic to produce large-scale shifts in global echinoid diversity patterns that we would be able to detect in our analysis. This provides yet another example of how trends in echinoid-targeting predatory drilling is distinct from what is observed in mollusks across the same interval. We have added some text to the Discussion section exploring this concept.

In response to more specific questions:

Is there any indication from the echinoids that larger individuals were drilled after the Eocene?

We have included a discussion of prey and predator relative sizes in the supplemental material. Though size data was collected while building the dataset, the sample size is not adequate to interpret any definitive patterns in drilled vs. undrilled echinoid sizes across this interval.

Do the authors have any data of urchin drilling from sites or times where cassids do not occur?

Our dataset includes data from Jurassic and Early Cretaceous echinoid populations, before the oldest known cassid fossil in the Late Cretaceous. We have expanded upon the discussion of echinoid-targeting drilling during this interval in the Discussion section.

I note from my experience that cassids are never very common, at least in the shallow-water habitats I have worked in. Can the authors give a percentage of the urchin species that were undrilled, and is there any indication of how those urchins might differ from those that were drilled frequently?

The low frequency of drilled echinoid populations, even in the Cenozoic, makes it difficult to systematically identify meaningful differences between drilled and undrilled species. One aspect that we do account for is differences between infaunal (irregular) and epifaunal (regular) echinoid species. Currently, preliminary data indicated that infaunal species are more frequently drilled than epifaunal species, though the notable underrepresentation of epifaunal taxa in the current iteration of the dataset makes it difficult to assess the observed difference in a rigorous statistical manner.

Referee: 2

Comments to the Author(s)

Overall, this is an interesting, well-written and well-argued paper that is of broad paleobiological interest. However, there is one critical issue that is problematic and most comments that follow are devoted to that.

These comments are based in part on some analyses of the data (S1) provided by the authors.

The problematic issue is the authors' combining two data sets into one, and performing quantitative analyses, central to the results of study, only on the combined data. The two data sets are (1) museum- and collections-derived data (EAT), and (2) literature-derived data (LIT). Each of the two sets provides some interesting insights into the timing of intensification of drilling predation on echinoids, perhaps complementary insights, but the data differ in substantive/significant ways (see Table Rev below). The authors offer no explicit discussion of (a) how these data differ from each other, (b) why they differ in these substantive ways, and, (c) why, given these significant differences, treating them together in the quantitative analyses is justified.

Table Rev: Some metrics for the two data sets and the combined data as calculated from data provided:
specimens pops drilled DFmean DFmed Czpops CZpops

	n	n	n	%				
Combined	34823	263	2841	0.08	0.01	186	0.71	
EAT	9315	201	321	0.02	0.00	125	0.62	
LIT	25508	62	2520	0.26	0.15	61	0.98	

The LIT and EAT data differ significantly from each other in almost every metric, especially in DFs (mean DFs differ by an order of magnitude). The differences in DFs between EAT and LIT may arise because LIT data are very much non-random with respect to DF, with researchers (with exception of Grun 2014) publishing results only for populations with DF>0, whereas EAT data are more or less random with respect to DF (presence/absence of drilled specimens, and DF values, was not a criterion for EAT collections, though it is possible that in some museum collections drilled specimens were removed by specialists for further study, although that seems highly unlikely and the authors would probably be aware of this). As a consequence, populations with DF=0, are counted (and very common, 123 of 201 pops) in EAT but only one instance is reported in LIT (Grun 2014).

Given the differences, should these data sets be combined for the analyses of the timing of drilling intensification? Statistically, I do not know how this can be justified. No procedure that I am aware of (resampling, differencing, etc.) can be used to somehow negate or equalize the very different biases of the two data sets. Even if the authors can answer these questions, it would be interesting to analyze EAT and LIT separately and see the congruence (or lack of congruence) in the results. In that context, the methods employed by the authors seem perfectly appropriate for EAT, and it is those analyses that would be the most persuasive because of the non-random nature of the EAT data. Treating the LIT and EAT independently and analyzing each separately, could strengthen the conclusions of the study, namely, that the increase in DF in echinoids was coincident with the diversification of cassids. If analyses of EAT and LIT produce different answers about the timing of the intensification, that would be of interest as well, as would a comparison of EAT and LIT results to the "combined" result.

We thank Referee 2 for their thorough inspection of the data, and agree that it could be potentially problematic that the literature dataset (LIT) skews towards populations with noticeably higher drilling frequencies than what is seen in the EAT dataset (see Fig. S1), perhaps due to the biased reporting of high population drilling frequencies in the literature. Alternatively, most of our literature data comes from Neogene and Quaternary populations, when drilling frequencies are expected to be high. In our initial analysis, the LIT dataset was incorporated into the final analysis

due to gaps and under sampled intervals in the original EAT dataset, mostly in the Miocene, Pliocene, and Pleistocene.

To determine the validity of our combined analysis, we re-analyzed the data using only the drilling frequencies from the EAT dataset, and find that the exclusion of the LIT data does not alter our results or interpretations. The Eocene-Oligocene interval was still identified as the most likely transitional interval in the three-phase models (Fig. S8) and the Eocene as the most likely intensification interval in the 2-phase models (Fig. S7). Notably, in the EAT-only three-phase simulations, the best fit model identified a transitional interval that is wider than but still overlaps the best-fit transitional interval identified using the combined EAT and LIT datasets (51.9 to 21.7 Ma vs. 44.5 Ma to 31 Ma). This suggests that exclusion of the literature data does not affect the identification of the most likely interval of drilling intensification but does moderately increase temporal uncertainty about the timing of the transition.

We have maintained the use of both datasets in the main text (for improved analytical resolution), and have provided the results of the EAT only analysis in the supplemental material. We believe this is justified. First, the analysis without the LIT data do not significantly change the results. Second, the bulk of the LIT data is from the Neogene and Quaternary (after the time period of interest for intensification of drilling in echinoids in the Eocene). We have also improved transparency as to the source of the data in both the main and supplemental manuscript in the following ways:

- We have re-run both two-phase and three-phase MCMC analyses using only the EAT dataset, and have found that the resulting best-fit model is consistent with the outcome of the combined EAT and LIT datasets. We have added three additional supplemental figures (Fig. S5, Fig. S7 and S8) to show first differences in drilling frequencies and the two- and three-phase model results using the EAT-only dataset, to be compared with the combined EAT and LIT analyses previously included.
- We have modified Figure 1 in the main manuscript so that population drilling frequencies from the LIT dataset are differentiated from EAT dataset populations.
- We have also clarified the sources of data in the Methods and Results sections of the main manuscript, and have modified figure captions in the main manuscript and supplemental material where appropriate to clarify the sources of the data shown.
- We have added a discussion to the Methods section regarding the potential bias of the literature data, and have illustrated the differences in population-level drilling frequency distributions between the two datasets in Figure S1.

In response to more specific questions:

Additional comments (italicized)

There are some minor discrepancies in the text of the ms.

Comment 1.

Text lines 85-89: the authors mentioned two separate tables Table S1 (EAT) and Table S2 (literature). I assume these were merged into Table S1 in the Supp. as Table S2 in Supp is something different

This was corrected in the text.

Comment 2.

Line 93: "A total of 263 populations (198 from EAT and 65 from literature sources)" 263 are listed in the S1 but 201 are EAT (not 198) and 62 LIT (not 65).

This was corrected in the text.

Comment 3.

Figure S1 – "Paleogeographic maps showing the geographic extent of EAT Database Project sampling" This map clearly includes data from LIT and EAT and not just the latter. This should be made explicit.

The figure caption has been edited.

Appendix B

Response to Referees:

Editor:

Please ensure data accessibility is made crystal clear in your MS-- from <https://www.eat-project.org/> it is still not immediately evident where the dataset and any associated code from this study is located and if it is 100% public.

The dataset has been uploaded to Dryad. DOI: <https://doi.org/10.5061/dryad.wstqjq2kn>

Referee: 1

Comments to the Author(s).

The authors have largely answered my previous queries and have cleared up some of their earlier confusions and misconceptions. I still would have liked a more granular analysis of the data, including instances where echinoids were not drilled and potential cases where cassids occur but their drill holes do not.

We have included a discussion in the supplemental material regarding exploring trends in echinoid drilling predation in a case-by-case sense, especially with consideration for the occurrence of cassid predators in these communities. The main difficulty workers face in this regard is the mismatched preservation potential of the prey and predator, with cassids and echinoids being very rarely preserved in the same location. As many of the echinoid populations occurring in the surveyed interval were also monospecific, this heterogeneity substantially limits our ability to glean larger temporal trends from a more granular treatment of the data.

Referee: 2

Comments to the Author(s).

Figure S1: The "numbering" on the panels (D = Cretaceous) and the caption (A=Cretaceous) is reversed.

We have corrected this in the supplemental.

Figure S2 caption. "This is likely due to both disproportionately greater reporting of highly drilled populations from the primary literature (ref. 1), and also potentially due to the literature data predominantly representing Neogene and Quaternary populations, when drilling frequency is expected to be relatively higher."

The above is undoubtedly correct but a few additional comments may be in order:

- a. Are the two distributions statistically different? (Chi Squared or some other test)
- b. The Lit data contain only a single population without drilled echinoids (Grun et al. 2014 Historical Biology), whereas in EAT, those predominate (>50%). So can the absence of such reports in the Lit data be biologically real? That's not likely. Therefore, not only does LIT appear biased towards the high frequencies, but it is also biased against 0 frequencies. I think this is worth pointing out, even if the MCMC simulations using both the combined EAT and LIT dataset as well as the EAT-only dataset produced similar results.

We have included the results of a chi-square test (the p-value) in the figure caption for S2, showing significant difference between the drilling frequencies of the EAT and LIT dataset. We have also included a discussion of the absence of data reporting a drilling frequency of 0 in the LIT dataset. We have added an additional discussion in the Supplemental Methods section.